# The Effect of Polarized Training (SIT, HIIT, and ET) on Muscle Thickness and Anaerobic Power in Trained Cyclists

**DOI:** 10.3390/ijerph18126547

**Published:** 2021-06-18

**Authors:** Paulina Hebisz, Rafał Hebisz

**Affiliations:** Department of Physiology and Biochemistry, University School of Physical Education in Wroclaw, 35 I.J. Paderewski Avenue, 51-612 Wroclaw, Poland; rafalhebisz@poczta.fm

**Keywords:** polarized training, sprint interval training, high intensity interval training, muscle thickness, anaerobic capacity, trained athletes

## Abstract

This study was undertaken to investigate the effect of two different concepts in a training program on muscle thickness and anaerobic power in trained cyclists. Twenty-six mountain bike cyclists participated in the study and were divided into an experimental group (E), which performed polarized training, comprising sprint interval training (SIT), high-intensity interval training (HIIT), and endurance training (ET), and a control group (C), which performed HIIT and ET. The experiment was conducted over the course of 9 weeks. Laboratory tests were performed immediately before and after the conducted experiment, including an ultrasound measurement of the quadriceps femoris muscle thickness and a sprint interval testing protocol (SITP). During the SITP, the cyclists performed 4 maximal repetitions, 30 s each, with a 90-s rest period between the repetitions. SITP was performed to measure maximal and mean anaerobic power. As a result of the applied training program, the muscle thickness decreased and the mean anaerobic power increased in the experimental group. By contrast, no significant changes were observed in the control group. In conclusion, a decrease in muscle thickness with a concomitant increase in mean anaerobic power resulting from the polarized training program is beneficial in mountain bike cycling.

## 1. Introduction

Athletes practicing endurance sports, such as cross-country running, cycling, and cross-country skiing, use different concepts in their training programs to improve athletic performance [1,2]. It has been proven that peak performance in endurance sports requires a high level of maximal oxygen uptake (VO_2_max), a common measure of aerobic capacity [3,4,5]. However, in the abovementioned sports disciplines, the racing effort is characterized by variable intensity, repeatedly reaching the maximal level of intensity. Therefore, it is increasingly important to develop not only aerobic but also anaerobic capacity [6]. Anaerobic capacity is the ability to perform short maximal efforts to achieve maximal anaerobic power [7]. According to some authors, the development of maximal anaerobic power is accompanied by an increase in muscle thickness [8]. Significant increases in muscle thickness and mass in sports such as cycling, cross-country running, and cross-country skiing are not beneficial, as competitions usually take place in mountainous or hilly terrains. Sports performance is affected by body mass in conditions such as those [9,10]. In light of the above, the effectiveness of various training programs concepts in endurance sports characterized by variable intensity is assessed.

One training program concept is the use of high-volume, low-intensity endurance training (ET) [1,11]. During ET trainings, the intensity is below 65% of maximal aerobic power as well as below the lactate threshold [12]. Another training concept is to combine ET trainings with those performed at or slightly above lactate threshold training (TT), at an intensity of 65–80% of maximal aerobic power [1,11,12,13,14]. The next concept is to use polarized training, which involves a combination of ET, TT, and interval training (IT) performed well above the lactate threshold and at intensities exceeding 80% of maximal aerobic power [1,11,12,13,14]. Two of the most common forms of IT are high-intensity interval training (HIIT) and sprint interval training (SIT), which differ in terms of the duration and intensity of repetitions, in addition to the duration of rest intervals between repetitions [15,16,17]. Typically, HIIT-type interval training is used in polarized training [1,18].

Due to the variable intensity of the racing effort, in the aforementioned sports disciplines, the concept of polarized training seems to be an optimal approach. This was supported by a study by Stöggl and Sperlich [1] conducted on endurance athletes, in which four training strategies were compared: (1) polarized training, (2) high-volume, low-intensity training, (3) threshold training, and (4) HIIT training, each used for 9 weeks. It was demonstrated that polarized training produces the best results in terms of developing peak oxygen uptake, peak power, and peak velocity and improves time to exhaustion during the ramp protocol [1]. Similarly, training cyclists showed greater improvements in peak power output, lactate threshold, and high-intensity exercise capacity [18], while recreational runners achieved improvement in 5-km performance and a greater increase in maximal oxygen uptake (VO_2_max) [19] as a result of polarized training, compared to other training strategies.

The studies cited in the foregoing paragraph on polarized programs used HIIT-type interval training [1,18,19]. According to some, polarized training involving simultaneous use of two types of interval training, HIIT and SIT, is beneficial in the development of aerobic capacity, as indicated by significant increases in VO_2_max and maximal aerobic power (APmax) in trained cyclists [17,20]. One may wonder what effect this combination of trainings would have on anaerobic capacity level and muscle thickness. The target intensity during HIIT is typically “near maximal”, or between 80% and 100% of APmax, with single periods of physical activity usually taking 4 min, while SIT protocols commonly include “all-out” efforts often taking 10–30 s [21,22]. SIT improves not only aerobic capacity indicators but also anaerobic capacity, as indicated by the increase in peak power and mean power during short maximal efforts [23,24]. Estes et al. [25] indicate that the use of interval training, in addition to the development of aerobic capacity, affects the increase in muscle mass and thickness of the vastus lateralis muscle among nonathletic adolescents. Naimo et al. [8] demonstrated an increase in peak and mean anaerobic power with a simultaneous increase in thigh muscle thickness under the influence of interval training among hockey players.

Having considered the foregoing, the aim of this study was to investigate whether in a group of trained cyclists, the use of polarized training, including two types of interval training, SIT and HIIT, as well as endurance training, affects the changes in quadriceps femoris muscle thickness, in addition to maximal and mean anaerobic power, measured during the sprint interval testing protocol (SITP).

## 2. Materials and Methods

### 2.1. Participants

The study involved twenty-six mountain bike cyclists (males). Each participant was characterized by at least three years of training experience in cycling. In addition, each participant declared the following with regard to the three-year period prior to entering the experiment: (1) training for at least 10 h per week (not including rest periods), (2) participating in a minimum of 15 cycling races per year, and (3) no regular (performed at least once every two weeks) trainings described as sprint interval training (SIT) below. The participants were randomly divided into two groups: control (C, *n* = 12) and experimental (E, *n* = 14). The characteristics of the groups are shown in Table 1.

The study design was approved by the Ethics Committee of the University School of Physical Education (Consent number: 39/2019) and carried out in accordance with the Declaration of Helsinki. Written informed consent was obtained from the participants and their guardians after the study details, procedures, benefits, and risks were explained.

### 2.2. Course of the Experiment 

Prior to the start of the experiment (the course of which is shown in Figure 1), each participant reduced their training volume to 2 sessions per week for a period of 6 weeks. During this period, cyclists were not involved in physical activities at intensities exceeding 70% maximal heart rate (HRmax). The experiment lasted 9 weeks and was conducted during the preparatory period. During the experiment, group E cyclists performed:-twice a week, sprint interval training (SIT), which consisted of 12–16 repetitions at maximal intensity, taking 30 s. The activity periods were divided into sets, and 4 repetitions were performed in each set. A low-intensity active rest of 90 s was used between repetitions, during which the power did not exceed 50 W. A 25-min active rest was used between sets, during which the first 2 min were performed at an intensity of 20% APmax, the next 20 min at approximately 50% APmax, and the last 3 min at 20% APmax. The course of the SIT is shown in Figure 2.-once a week, high intensity interval training (HIIT), which included several (5–7) 5-min activity periods at an intensity of 85–95% APmax, interspersed with a 12-min activity at 50% APmax.-twice a week, endurance training (ET), performed at an intensity of 55–60% APmax, for a total of 120–180 min.

As a part of the experiment, group C cyclists performed:
-twice a week, HIIT training (trainings were performed as described above, the same as in group E).-three times per week, ET training (trainings were performed as described above, the same as in group E).

Two days per week were dedicated to active or passive recreation in each group. The total weekly training volume was 10–13 h for each participant in the experiment and did not differ significantly between the groups. In the fifth week of the experiment, a recovery cycle was applied, featuring 50% less total training volume in both groups (5–6.5 h), with no change in training methods or intensity.

During training, power output was monitored using a PowerTap G3 ANT+ and GS ANT+ system (PowerTap, Madison, WI, USA), whereas heart rate (HR) was monitored using a Garmin Edge 520 and Edge 810 system (Garmin Ltd., Olathe, KS, USA).

### 2.3. Study Procedures

Laboratory testing was performed immediately before and after the experiment, which included an incremental testing protocol (ITP), ultrasound measurement of quadriceps femoris muscle thickness, and performing the sprint interval testing protocol. During the 24 h that preceded the exercise tests, the participants did not perform any training. All the foregoing tests and measurements were performed in controlled laboratory conditions (temperature and humidity controlled) at the Exercise Laboratory of the University School of Physical Education (PN-EN ISO 9001:2001 certified).

#### 2.3.1. Incremental Testing Protocol (ITP)

The test was performed on a Lode Excalibur Sport cycle-ergometer (Lode BV, Groningen, The Netherlands), which was calibrated before the tests. The physical activity started with a load of 50 W, and every 3 min the load was increased by 50 W, until the participant refused to continue. If a participant was unable to complete an entire 3 min stage, 0.28 W per second missed was subtracted from the work rate at that stage. The highest power output determined in the ITP was taken to be the measure of maximal aerobic power (APmax).

Respiratory function was measured during the test. The cyclist wore a mask connected to a Quark gas analyzer (Cosmed, Milan, Italy). The gas analyzer was calibrated before use with a reference gas mixture of carbon dioxide—5%, oxygen—16%, and nitrogen—79%. Respiratory parameters were measured in each recorded breath (breath-by-breath) and then averaged over 30-s intervals. The maximal oxygen uptake (VO_2_max) was determined from the recorded data.

#### 2.3.2. Sprint Interval Testing Protocol (SITP)

The test was also performed on a Lode Excalibur Sport cycle-ergometer, which was calibrated before the tests. The test was preceded by a 20-min warm-up, during which the participant exercised at an intensity of 40% APmax (as determined by the ITP) for 5 min, and then at 60% APmax for 15 min. A low-intensity active rest of 10 min was used after the warm-up. This was followed by 4 maximal repetitions—30 s each, during which the participant was supposed to achieve as much power as possible in the shortest time and maintain it for as long as possible. An active rest period of 90 s was used between the repetitions, during which the participant pedaled at 30 W.

The power was measured during each repetition. Maximal and mean anaerobic power measured during the best repetition (P_max_ and P_mean1_) and mean anaerobic power determined from all four repetitions (P_mean4_) were used in the data analysis.

#### 2.3.3. Muscle Thickness Measurements (MT)

Prior to the sprint interval testing protocol, participants’ quadriceps femoris muscle thickness and body mass were measured using a BodyMetrix™ System ultrasound device (Hosand Technologies, Verbania, Italy) with an A-mode probe at 2.5 MHz. This device was validated for tissue thickness measurements by Ribeiro et al. [26]. A layer of ultrasound gel was applied to the participant’s skin surface before measurement. The right and left quadriceps femoris muscle thickness was measured after ultrasound imaging, along the course of the rectus femoris muscle. The measurement was done in a standing position. The total thickness of the muscle layer along the measurement taken was used in the data analysis, comprising the rectus femoris layer and the vastus intermedius layer. The site where the muscle thickness was the largest was used for analysis; the result was recorded in millimeters. The same person performed the measurement both before and after the experiment in all participants.

Additional measurements were taken to assess the reliability of the quadriceps femoris muscle thickness measurement procedure used. A randomly selected twenty-nine students (age: 21.3 ± 0.6 [years]; body mass: 70.3 ± 9.8 [kg]; body height: 176.6 ± 8.9 [cm]) of the University School of Physical Education participated in the measurements. The right and left quadriceps femoris muscle thickness was measured twice in each of the students. The measurement was performed as described above using the same a BodyMetrix™ System ultrasound device. (Hosand Technologies, Verbania, Italy). In the group of students, the measurement was made by the same person who performed the measurement among the cyclists.

### 2.4. Statistical Analysis

Statistica 13.1 software was used for statistical calculations. Arithmetic mean and standard deviation were calculated. Analysis of variance with repeated measures and Scheffe’s post hoc test were used to determine whether there were statistically significant differences in the parameters evaluated between groups E and C, as well as between tests performed before and after the experiment. The *p* < 0.05 level was taken as statistically significant. 

In order to assess the reliability of the muscle thickness measurement, the Cronbach’s alpha coefficient (*α*) was calculated. Then, the value of standard error measurement (*SEM*) was calculated based on the following formula:(1)$$SEM=SD\cdot \sqrt{1-\alpha }$$

## 3. Results

Analysis of variance demonstrated statistically significant main effects for interactions between groups and repeated measures in terms of the right quadriceps femoris muscle thickness (F = 8.68; *p* = 0.007; η^2^ = 0.27), left quadriceps femoris muscle thickness (F = 14.80; *p* = 0.000; η^2^ = 0.38), mean anaerobic power determined from all four repetitions (P_mean4_) of the sprint interval testing protocol (F = 8.67; *p* = 0.007; η^2^ = 0.27) (Table 2), maximal oxygen uptake (F = 8.30; *p* = 0.008; η^2^ = 0.27), and maximal aerobic power (F = 6.69; *p* = 0.017; η^2^ = 0.23) (Table 1). Based on post hoc tests, the thickness of the right and left quadriceps femoris muscles decreased statistically significantly in the E group. In addition, P_mean4_ increased significantly in the E group. In contrast, no significant changes in muscle thickness or anaerobic power values were observed in the C group (Table 2). Maximal oxygen uptake and maximal aerobic power increased significantly only in the E group (Table 1).

An additional measurement of the reliability of the procedure among students showed that α for the left and right quadriceps femoris muscle thickness was 0.995 and 0.994, respectively. In turn, the SEM for the left and right quadriceps femoris muscle thickness was 0.507 mm and 0.542 mm, respectively.

## 4. Discussion

In the presented study, it was shown that in a group of trained cyclists, under the influence of applying polarized training, including endurance training and two types of interval training, SIT and HIIT, the quadriceps femoris muscle thickness decreased. Concurrently, there was an increase in mean anaerobic power determined from four repetitions performed during the sprint interval testing protocol, while the value of maximal anaerobic power did not change. These results may complement previous studies that described the beneficial effects of polarized training on aerobic capacity levels in endurance athletes [17,20,27]. As in the presented study, among cyclists of the experimental group, an increase in the maximal oxygen uptake and maximal aerobic power was additionally observed.

The decrease in the quadriceps femoris muscle thickness in the presented study is a different effect compared to the results of studies published in other scientific articles. Numerous authors conclude that undertaking regular strength [28,29,30,31], endurance [32,33], and interval [25] physical activity increases muscle thickness. An increase in muscle thickness has also been observed as a result of training programs involving strength training and endurance training [30,31,33], as well as programs comprising strength training and interval training [25,34]. The study participants in the foregoing articles were nontraining individuals. In contrast, the study described in this manuscript involved cyclists who systematically trained prior to the experiment. In view of this, the initial level of physical activity can be considered to be a factor explaining the differences between the findings of the present study and those of the authors cited above.

Among endurance athletes, the implementation of strength training may be a stimulus to provoke an increase in muscle thickness [35]. The implementation of SIT-type training among trained hockey players had a similar effect [8]. Nevertheless, several authors have argued that the use of training programs created using strength training and endurance training does not lead to an increase in muscle thickness among athletes [36,37,38]. Strength training was not performed in our experiment, yet it did include SIT along with HIIT and endurance training, similar to the study by Naimo et al. [8]. In our experiment, performing endurance training may have been a factor that inhibited the development of muscle thickness that we expected to result from intensive SIT training among mountain bike cyclists. Moreover, in our experiment, we observed a decrease in the thickness of the right and left quadriceps femoris muscles in the E group, which was a surprising effect. It is possible that differences in the effects of the experiment described in this manuscript and the experiment by Naimo et al. [8] may also result from a different SIT training protocol. Our study involved maximal repetitions taking 30 s each. Naimo et al. [8] also used maximal repetitions lasting 10–30 s. The most important difference concerned the rest periods separating the repetitions. In our study, we used an active rest of 90 s between maximal sprints, whereas the rest period in the study by Naimo et al. [8] was 4 min. Campos et al. [39] showed that using a short rest period between repetitions, which does not enable optimal rest, and performing a high number of repetitions during strength training do not affect changes in muscle thickness. The opposite effect, i.e., an increase in muscle thickness, was obtained in strength training with long rest intervals and low repetition numbers [39]. Other studies (on animals) have shown that the use of intervals during resistance training [40] or interval training [41] that do not facilitate optimal rest between repetitions results in muscle thickness reduction (muscle atrophy). This is because the application of a short rest period resulted in an increased expression of proteins with a catabolic function and a decrease in expression of proteins with an anabolic function [40].

In the presented study, it was impossible to explain the mechanisms of the observed muscle atrophy because no muscle biopsy was performed. Anoveros-Barrera et al. [42] indicated that muscle biopsy allows the assessment of muscle for morphological, cellular, and biochemical features. Biopsy can be assessed for the expression of genes and their involvement in catabolic and anabolic pathways of muscle, which were mentioned above [40,42]. According to some authors, the changes in muscle thickness may be the result of the change in the angle between the longitudinal axis of the entire muscle and its fibers, defined as the pennation angle [43,44]. The muscle hypertrophy and greater muscle tone involve an increase in fiber pennation angles, while a greater value of the fiber pennation angles results in a decrease in muscle strength [44]. In the presented study, muscle strength was not assessed, but the maximal anaerobic power was measured, which did not change. Therefore, most likely the pennation angle did not change either. However, the degree of fiber pennation angles can be detected by ultrasonic investigation from the image of a B-mode ultrasonogram, which we did not perform. The lack of muscle biopsy and measurement of fiber pennation angles in the present study is a limitation and indicates a direction for further detailed research.

The foregoing changes in the muscle thickness may be important in sports competitions, as numerous authors report that athletic performance in mountain bike cycling is correlated with power at metabolic thresholds [9], as well as maximal power of incremental tests [10] expressed in W∙kg^−1^. Furthermore, Bejder et al. [45] and Inoue et al. [46] demonstrated that mean power (expressed in W∙kg^−1^) in a sprint interval training consisting of four sprints of 30 s each is strongly correlated with the athletic performance of mountain bike cyclists. The authors’ previous original studies have shown that polarized training leads to an increase in maximal power in the incremental testing protocol [20,27]. The results of the present study proved an increase in mean power in a sprint interval testing protocol. The foregoing effects were obtained in the absence of significant changes in body mass. Having considered that, these effects can positively impact sport performance in mountain bike cycling, as they lead to improved power measured in W∙kg^−1^.

## 5. Conclusions

As a result of the polarized training program, the quadriceps femoris muscle thickness decreased, accompanied by an increase in the mean anaerobic power of the sprint interval testing protocol. Such changes are beneficial to achieving good sport performance in mountain bike cycling.

## Figures and Tables

**Figure 1 ijerph-18-06547-f001:**
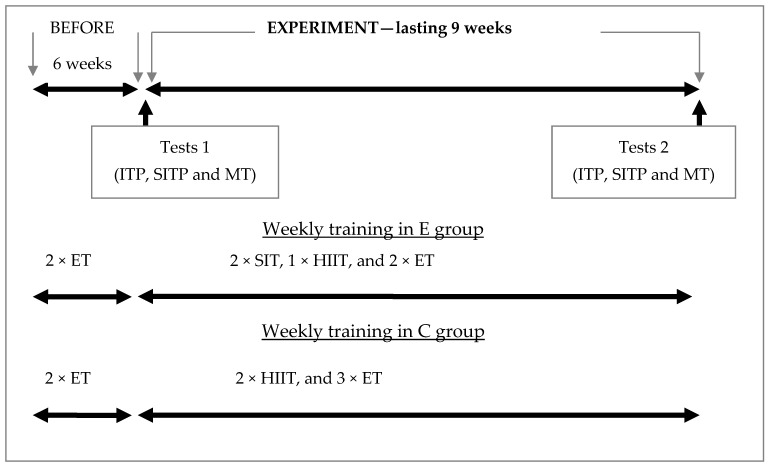
Scheme of the experiment (ITP—incremental testing protocol; SITP—sprint interval testing protocol; MT—muscle thickness measurements; E—experimental group; C—control group; ET—endurance training; SIT—sprint interval training; HIIT—high intensity interval training).

**Figure 2 ijerph-18-06547-f002:**
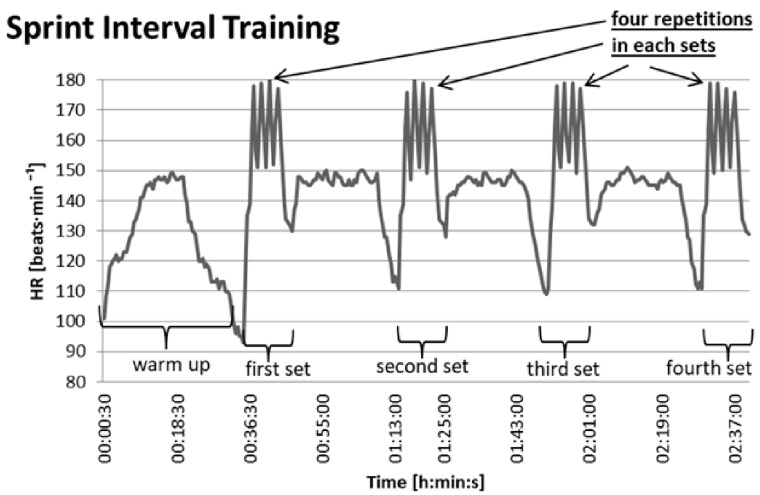
Diagram representing the sprint interval training with the heart rate (HR) of one of the participants.

**Table 1 ijerph-18-06547-t001:** Characteristics of the cyclist groups studied.

Group	VO_2_max [mL^−1^·min^−1^·kg]	APmax [W]	Age [years]	Body Mass [kg]	Body Height [cm]
**Pre-Experiment**					
E	62.3 ± 6.4	380.8 ± 50.0	21.7 ± 7.7	69.6 ± 8.1	179.1 ± 6.2
C	59.6 ± 8.4	369.6 ± 29.2	20.5 ± 5.5	70.2 ± 8.9	177.5 ± 6.1
**Post-Experiment**					
E	66.3 ± 5.6 *	398.3 ± 37.8 *	21.7 ± 7.7	69.9 ± 7.6	179.1 ± 6.2
C	61.2 ± 10.1	373.8 ± 29.1	20.5 ± 5.5	69.4 ± 9.1	177.5 ± 6.1

E—experimental group; C—control group; VO_2_max—maximal oxygen uptake measured during the incremental testing protocol; APmax—maximal aerobic power measured during the incremental testing protocol; *—*p* < 0.05—significant difference between pre- and post-experiment value; data are presented as mean ± standard deviation.

**Table 2 ijerph-18-06547-t002:** Pre- and post-experiment muscle thickness and anaerobic power measures.

	Pre-Experiment			Post-Experiment		
	Mean ± SD	95% CI Lower Upper		Mean ± SD	95% CI Lower Upper	
**Experimental group**						
MT-r (mm)	54.14 ± 5.4	51.02	57.26	50.28 ± 6.03 *	46.8	53.76
MT-l (mm)	50.5 ± 6.34	46.83	54.16	46.14 ± 6.06 *	42.64	49.64
BM (kg)	69.67 ± 8.07	65.01	74.33	70.21 ± 8.37	65.37	75.05
P_max_ (W)	1216.2 ± 253.7	1069.7	1362.7	1220.3 ± 239.2	1082.17	1358.42
P_mean1_ (W)	726.3 ± 86.39	676.41	776.18	736.0 ± 101.7	677.29	794.76
P_mean4_ (W)	626.41 ± 75.54	582.79	670.03	648.69± 78.7 *	603.24	694.13
**Control group**						
MT-r (mm)	49.0 ± 5.25	45.66	52.34	50.58 ± 5.96	46.79	54.37
MT-l (mm)	46.08 ± 6.12	42.19	49.97	48.33 ± 6.52	44.18	52.48
BM (kg)	70.25 ± 8.91	64.58	75.91	69.75 ± 8.65	64.25	75.25
P_max_ (W)	1301.8 ± 235.8	1151.9	1451.7	1271.9 ± 230.0	1125.83	1418.13
P_mean1_ (W)	721.82 ± 77.17	672.79	770.85	719.02 ± 94.26	659.13	778.92
P_mean4_ (W)	610.19 ± 61.57	571.06	649.31	619.25 ± 50.84	586.94	651.55

MT-r—right quadriceps femoris muscle thickness; MT-l—left quadriceps femoris muscle thickness; BM—body mass; P_max_—maximal anaerobic power reached during SITP; P_mean1_—mean anaerobic power reached during the repetition with the highest P_max_; P_mean4_—mean anaerobic power calculated from performing all four repetitions during SITP; *—*p* < 0.05—significant difference between pre- and post-experiment value.

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
