# Peer review of "The Effect of Polarized Training (SIT, HIIT, and ET) on Muscle Thickness and Anaerobic Power in Trained Cyclists"

_ijerph, 2021, doi:10.3390/ijerph18126547_

Round 1
Reviewer 1 Report
In this work, Hebisz &Hebisz propose a special program (sprint interval training or SIT)) for training cyclist to improve their athletic performance in terms of anaerobic performance without the undesirable hypertrophy effect on quadriceps femoris.
The experimental design overall is well conducted, although some considerations could be taken into account to improve the clarity of results of this exploratory study:
a) Although data presented in table 2, do not show notorious differences between experimental and control individuals previous and at the end of experiments, may be would be desirable applying an aleatorization process.
b) A blinded analysis must be considered, specially in quadriceps measurements by ultrasound (an operator-dependent variable).
The presented results would be of interest to the sports community.
Author Response
Reviewer 1
Comments and Suggestions for Authors
In this work, Hebisz &Hebisz propose a special program (sprint interval training or SIT)) for training cyclist to improve their athletic performance in terms of anaerobic performance without the undesirable hypertrophy effect on quadriceps femoris.
- Thank you very much for your review and the valuable comments.
The experimental design overall is well conducted, although some considerations could be taken into account to improve the clarity of results of this exploratory study:
- a) Although data presented in table 2, do not show notorious differences between experimental and control individuals previous and at the end of experiments, may be would be desirable applying an aleatorization process.
- b) A blinded analysis must be considered, specially in quadriceps measurements by ultrasound (an operator-dependent variable).
- We agree and this was corrected.
Additional measurements were taken to assess the reliability of the quadriceps femoris muscle thickness measurement procedure used. Randomly selected twenty-nine students, of the University School of Physical Education, participated in the measurements.
Such information and obtained results have been added to the Materials and Methods section and the Results section.
The presented results would be of interest to the sports community.
Reviewer 2 Report
I read Paulina Hebisz and Rafał Hebisz's manuscript entitled ' The Effect of Polarized Training (SIT, HIIT, and ET) on Muscle Thickness and Anaerobic Power in Trained Cyclists '.
The manuscript describes experimental research carried out on a sample of 26 mountain bike cyclists trained for at least three years to a group of which a polarized training was administered, i.e. consisting of endurance training, threshold training and interval training. Interval training in turn consisted of high-intensity interval training and sprint interval training.
The purpose of the study was to observe the effects of this type of training on anaerobic strength and muscle thickness.
The authors noted that the protocols used increased the average anaerobic strength and reduced the thickness of the femoral quadriceps muscle.
The increase in anaerobic strength complements previous studies in which an increase in aerobic capacity in the same type of athletes was also observed. Indeed it is not shown if this group of athletes reached an increased aerobic capacity after polarized training as previously demonstrated.
The decrease in the quadriceps femoris muscle thickness in the presented study is a different effect, compared to the results of studies published in other scientific articles and it is ascribed to the fact that, compared to previous studies, athletes trained for a long time were observed and different rest intervals have been studied that may have favoured catabolic protein processes.
Such changes are beneficial to achieving good sport performance in mountain bike cycling.
The experiment described is interesting, although it seems to be a series of previous research on the same subject published by the same authors, which, in my opinion, would have been more useful to discuss scientifically all together in a single work.
It is also unclear whether the data on the increase in average anaerobic strength (which probably had to be discussed together with the data on aerobic capacity obtained with the same type of polarized training) or the data on the reduction of the thickness of the quadriceps in the type of athletes studied are more interesting. In any case, in my opinion, these two aspects and their importance for performance in mountain bike-cycling and related sports should be emphasized in the introduction, as the main problem.
The reduction in the thickness of the quadriceps muscle is clearly a particular result and should be deepened. For example, it does not refer to the degree of pennation of muscle fibres that can be detected by ultrasonic investigation (Strasser EM et al. Association between ultrasound measurements of muscle thickness, pennation angle, echogenicity and skeletal muscle strength in the elderly. Age (Dordr). 2013;35(6):2377-2388. doi:10.1007/s11357-013-9517-z
On the other hand, the hypothesis of an increase in catabolic processes remains such in the absence of a muscular biopsy (Anoveros-Barrera A et al.Clinical and biological characterization of skeletal muscle tissue biopsies of surgical cancer patients. J Cachexia Sarcopenia Muscle. 2019 Dec;10(6):1356-1377. doi: 10.1002/jcsm.12466.
Experiments are described as results. Statistical evaluation of the data seems appropriate.
I, therefore, recommend accepting the manuscript for publication after minor revisions.
Author Response
Reviewer 2
Comments and Suggestions for Authors
I read Paulina Hebisz and Rafał Hebisz's manuscript entitled ' The Effect of Polarized Training (SIT, HIIT, and ET) on Muscle Thickness and Anaerobic Power in Trained Cyclists '.
The manuscript describes experimental research carried out on a sample of 26 mountain bike cyclists trained for at least three years to a group of which a polarized training was administered, i.e. consisting of endurance training, threshold training and interval training. Interval training in turn consisted of high-intensity interval training and sprint interval training.
The purpose of the study was to observe the effects of this type of training on anaerobic strength and muscle thickness.
The authors noted that the protocols used increased the average anaerobic strength and reduced the thickness of the femoral quadriceps muscle.
- Thank you very much for your review and the valuable comments.
The increase in anaerobic strength complements previous studies in which an increase in aerobic capacity in the same type of athletes was also observed. Indeed it is not shown if this group of athletes reached an increased aerobic capacity after polarized training as previously demonstrated.
- Yes, this group of athletes reached an increase aerobic capacity. In the first version of the manuscript, we deliberately didn't present this, because that wasn't the purpose of the study. Aerobic capacity was the subject our previous articles, and here we wanted to avoid it.
In the current version of the manuscript, we have added data showing improvement in aerobic capacity to the Results section and to Table 1. We mentioned about this change in the Discussion section.
The decrease in the quadriceps femoris muscle thickness in the presented study is a different effect, compared to the results of studies published in other scientific articles and it is ascribed to the fact that, compared to previous studies, athletes trained for a long time were observed and different rest intervals have been studied that may have favoured catabolic protein processes.
Such changes are beneficial to achieving good sport performance in mountain bike cycling.
The experiment described is interesting, although it seems to be a series of previous research on the same subject published by the same authors, which, in my opinion, would have been more useful to discuss scientifically all together in a single work.
- Indeed, this experiment is a series of related studies. It would be interesting, but possible as a review, to describe in a single article the effects of polarized training on all the factors we have examined so far. ​In research articles, we try not to describe many factors, as we were previously negatively assessed by the Reviewers
It is also unclear whether the data on the increase in average anaerobic strength (which probably had to be discussed together with the data on aerobic capacity obtained with the same type of polarized training) or the data on the reduction of the thickness of the quadriceps in the type of athletes studied are more interesting. In any case, in my opinion, these two aspects and their importance for performance in mountain bike-cycling and related sports should be emphasized in the introduction, as the main problem.
- The Introduction section has been corrected.
The reduction in the thickness of the quadriceps muscle is clearly a particular result and should be deepened. For example, it does not refer to the degree of pennation of muscle fibres that can be detected by ultrasonic investigation (Strasser EM et al. Association between ultrasound measurements of muscle thickness, pennation angle, echogenicity and skeletal muscle strength in the elderly. Age (Dordr). 2013;35(6):2377-2388. doi:10.1007/s11357-013-9517-z
On the other hand, the hypothesis of an increase in catabolic processes remains such in the absence of a muscular biopsy (Anoveros-Barrera A et al.Clinical and biological characterization of skeletal muscle tissue biopsies of surgical cancer patients. J Cachexia Sarcopenia Muscle. 2019 Dec;10(6):1356-1377. doi: 10.1002/jcsm.12466.
- The indicated articles were used to improve the Discussion section.
Experiments are described as results. Statistical evaluation of the data seems appropriate.
I, therefore, recommend accepting the manuscript for publication after minor revisions.
Round 2
Reviewer 1 Report
The authors responded adequately to all the suggestions.